# Predictors of educational failure at 16 and 19 years of age—SESBiC longitudinal study

**Marie Bladh**[1]*, **Carl Göran Svedin**[2], **Sara Agnafors**[1]

**1** Division of Children's and Women's health, Department of Biomedical and Clinical Sciences, Linköping University, Linköping, Sweden, **2** Department of Social Science, Marie Cederschiöld University, Stockholm, Sweden

* marie.bladh@liu.se

**Data Availability Statement:** The data in the current study can, due to the legal agreement accepted upon delivery of data from the National Board of Health and Welfare, only be shared on a group level upon request by a researcher. For

## Abstract

### Background

Educational attainment is highly associated with future health and independence. Throughout childhood, children are exposed to factors that may promote educational attainment and factors that may be associated with a reduced likelihood of being able to complete their education. The purpose of the current study was to investigate which factors, measured from birth up to finishing upper secondary school, were associated with a lower mean grade point average from lower and upper secondary school as well as eligibility to upper secondary school and college/university.

### Methods

This is a longitudinal study on 1723 children born in 1995/1996 who have been followed until they were 20 years old. Information with respect to maternal sociodemographics, maternal stress factors during pregnancy and childhood, birth characteristics of the child, child behavior at 3 and 12 years of age, and mean grade point average from lower and upper secondary school, including eligibility to upper secondary school and college/university was collected.

### Results

Children exhibiting high problems scores on the child behavior checklist at 12 years of age and children or having other living arrangements (e.g. foster parents or institutional care) were less likely to fulfill the requirements for upper secondary school (OR = 0.35, 95% CI = 0.17–0.71 and OR = 0.33 95% CI = 0.17–0.65, respectively). The likelihood of fulfilling the requirements to college/university was lower if the child had divorced parents at three years of age (OR = 0.30, 95% CI = 0.16–0.58) and exhibited externalizing problems at 12 years of age (OR = 0.45, 95% CI = 0.24–0.86) and if the mother had experienced high level of stress at (OR = 0.32, 95% CI = 0.14–0.77).

### Conclusion

Identifying mothers with high level of stressors as well as children with externalizing behaviour problems to provide guidance and support is very important as these two factors

requests contact M. Bladh, MD or G. Sydsjö,
Professor, gunilla.sydsjo@regionostergotland.se."

**Funding:** Funding for this project was provided by
Skandia research.

**Competing interests:** The authors have declared
that no competing interests exist.

appear to be associated with future study performance in both lower and upper secondary
school.

## Introduction

Being able to complete primary and secondary education can be regarded as an essential
milestone for future independence, finding employment, and can be viewed as a predictor
of long-term achievement [1]. Studies have shown that students who fail to complete their
scholastic education have an increased risk for later mental disorders [2], unemployment,
and societal alienation. On the other hand socially disadvantageous environment during
childhood and adolescence might also have an impact on scholastic performance [3]. Sub-
stantial research has been performed with respect to which factors during childhood affect
academic achievements, including gene-environment interactions, birth characteristics,
upbringing, parental socio-economic status, school environment, and behavioral problems
[4].

Studies have shown that birth characteristics such as being born small for gestational age
(SGA), or with a (very) low birthweight (LBW), and prematurity are associated with a reduced
cognitive ability as well as neurodevelopment impairment [5]. A recent study by Lindström
et al. (2017) found SGA to be strongly associated with academic performance in grade 9 (age
16), the smaller the child was the more it affected the academic performance [6]. Regarding
children born large for gestational age (LGA), results have, on the contrary, shown similar or
better school performance compared to children born appropriate for gestational age (AGA)
[7, 8]. Preterm birth, before the 37th week of pregnancy, has been shown to be negatively asso-
ciated with school performance when controlling for socioeconomic factors [9]. Interestingly,
Ahlsson et al. (2015) also showed, that when restricting the analyses to siblings, only preterm
birth before 30 weeks of gestation remained significant [9]. Similarly, a negative association
between very low birthweight (VLBW) and reading skills have previously been found [10].
Likewise, children born with extremely low birthweight (ELBW) are at risk for the need for
special education and lower IQ scores in adolescence [11].

Additionally, in a large Scottish study comprising over 750,000 children, it was found that
children with a low five-minute-Apgar score were less likely to achieve a higher educational
level [12]. This finding is in line with a previous study of term singletons, where children with
a lower five-minute-Apgar score were found to be less likely to finish upper secondary school,
measured as receiving final grades in school year nine [13]. Several studies have also indicated
that maternal smoking during pregnancy may have long-term negative effect on the child's
educational achievement [14–17].

Previous research has found that psycho-social factors as well as behavioral factors affect
the completion of primary and secondary school, as well as through college/university [18].
Psychosocial stress and stressful life events during early life affect child development and
behaviour [19, 20]. Even prenatal and early life stress seem to have long lasting impact on child
development and well-being. In an Australian population based study, maternal life stress
events during pregnancy was found to have different effects on boys and girls [21]. Multiple
life stress events were associated with lower reading scores in girls, but higher reading and
mathematics scores in boys [21]. However, stressful life events potentially impact functioning
and wellbeing during all of childhood as shown by Dupéré et al. [22]. Moreover, socioeco-
nomic factors e.g. poverty has been linked to exposure of cumulative risk exposure in early

adolescence [23]. Thus, early life stress might be detrimental for child development through different pathways.

Both internalizing problems and externalizing problems have been shown to negatively influence educational achievement, by increasing the risk for dropping out of school [24] or reducing the number of years of schooling, i.e. settling for a lower level of educational attainment without dropping out of school and also lower grades and thus becoming non-eligible to higher education.

Few studies have had the opportunity to follow children during their entire childhood and through primary and secondary schooling and study the effects of early life risk factors and evaluate performance/achievements in lower and upper secondary school and its potential effect on eligibility to higher education.

Thus, the primary aim of the study was to investigate the academic transition between 16 years of age to 19 years of age in a Swedish setting. Secondary aims were to investigate factors affecting the academic achievements at both time points, focusing on birth characteristics, maternal social and mental status postnatally, socio-demographic and behavioral problems at 3 and 12 years of age.

## Methods

### Swedish school system—Primary and lower secondary schooling

The Swedish school system is based on 9 years of compulsory schooling, of which the last three years are equivalent to lower secondary schooling. After the 9th grade the student can choose to apply to a program-oriented education where the major programs are social sciences, vocational educations, sciences, and arts. In order to be accepted into one of these programs the student needs to make a formal application to one of the high schools, where acceptance is based on the grade point average and whether they fulfill the requirement of passable grades in Swedish, Mathematics, and English for all educations except sciences were a passable grade in natural sciences (which includes; physics, chemistry, and biology) is also required (Fig 1). Grade point average (GPA) after completion of the 9th grade ranges between 0 and 340, and are based on the student's best 16 (out of 17) grades.

### Swedish school system—Upper secondary schooling

Upper secondary schooling in Sweden, equivalent to grades 10–12, is not mandatory. However, most adolescents still apply and start a program in "high school" due to the expectations from society. Approximately 98–99% of the students who graduated from lower secondary school applied and were enrolled in a upper secondary school program (https://www.skolverket.se/download/18.6bfaca41169863e6a65a470/1553964853546/pdf2979.pdf). Students who did not fulfill the requirements for starting in a program after completion of lower secondary schooling (9th grade) have the opportunity to begin with an "introductory year" or "individual program" and then transfer to one of the standard three-year programs. Each year the student takes a number of classes, specific to their educational program in addition to Swedish, History, Social Science, English, and Mathematics which are classes that everyone takes. The only difference in these common classes are the difficulty levels in each program. The GPA after secondary schooling ranges between 0 and 22.5 (detailed information on how GPA is calculated can be found at: https://www.gymnasieguiden.se/informeras/meritpoaeng-meritkurser-omradeskurser).

English "map" explaining the Swedish school system: https://www.skolverket.se/download/18.6011fe501629fd150a2a50a/1532675074943/Karta_over_utbildningssystemet_eng.pdf

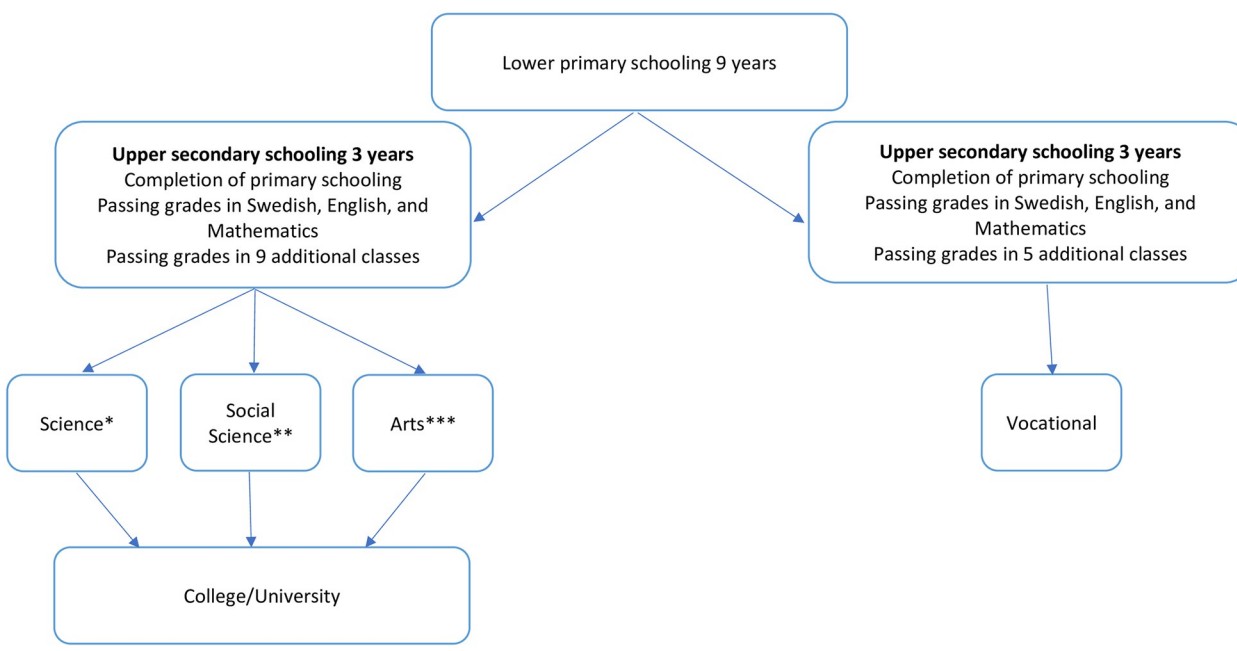

* Three of the additional classes have to be biology, physics, and chemistry
** Four of the additional classes have to be geography, history, social science, and religion
*** The additional courses can be any of the classes offered in primary schooling

**Fig 1. The Swedish school system.**

## Study population

The South East Sweden Birth Cohort study (SESBiC-study) is a longitudinal study comprising children (and their parents) born between May 1st 1995 and December 31st 1996. All mothers of children born between these dates in five adjacent municipalities, both urban and rural, in the south of Sweden were invited to participate. In total 1958 mohters were invited of which 1723 (88%) accepted participation, Fig 2. In the current study, 703 out of 1687 eligible children participated (2 children 4 mothers deceased, 10 moved out of the county and 24 intellectually disabled) at all follow-ups.

## Data collection

**Baseline.** Baseline information was retrieved at child welfare centers when the mothers visited the centers for the routine 3-month check-up of the child. During this check-up, the staff informed the parents regarding the study, parents who accepted participation gave their informed consent for participation to the welfare center staff.

**3-year follow-up.** The 3-year follow-up took place at the routine 3-year check-up at the child welfare center. The mother of the child was asked to fill out several questionnaires, where the questions covered areas of maternal life events and sense of coherence, having been through a divorce, as well as temperament and different scales pertaining to the health and behavior of the child.

**12-year follow up.** The 12-year follow-up was directed to both of the parents as well the child. A letter containing information on the upcoming follow-up with an invitation to participate were sent to the parents. A simplified information letter was enclosed to the children. If

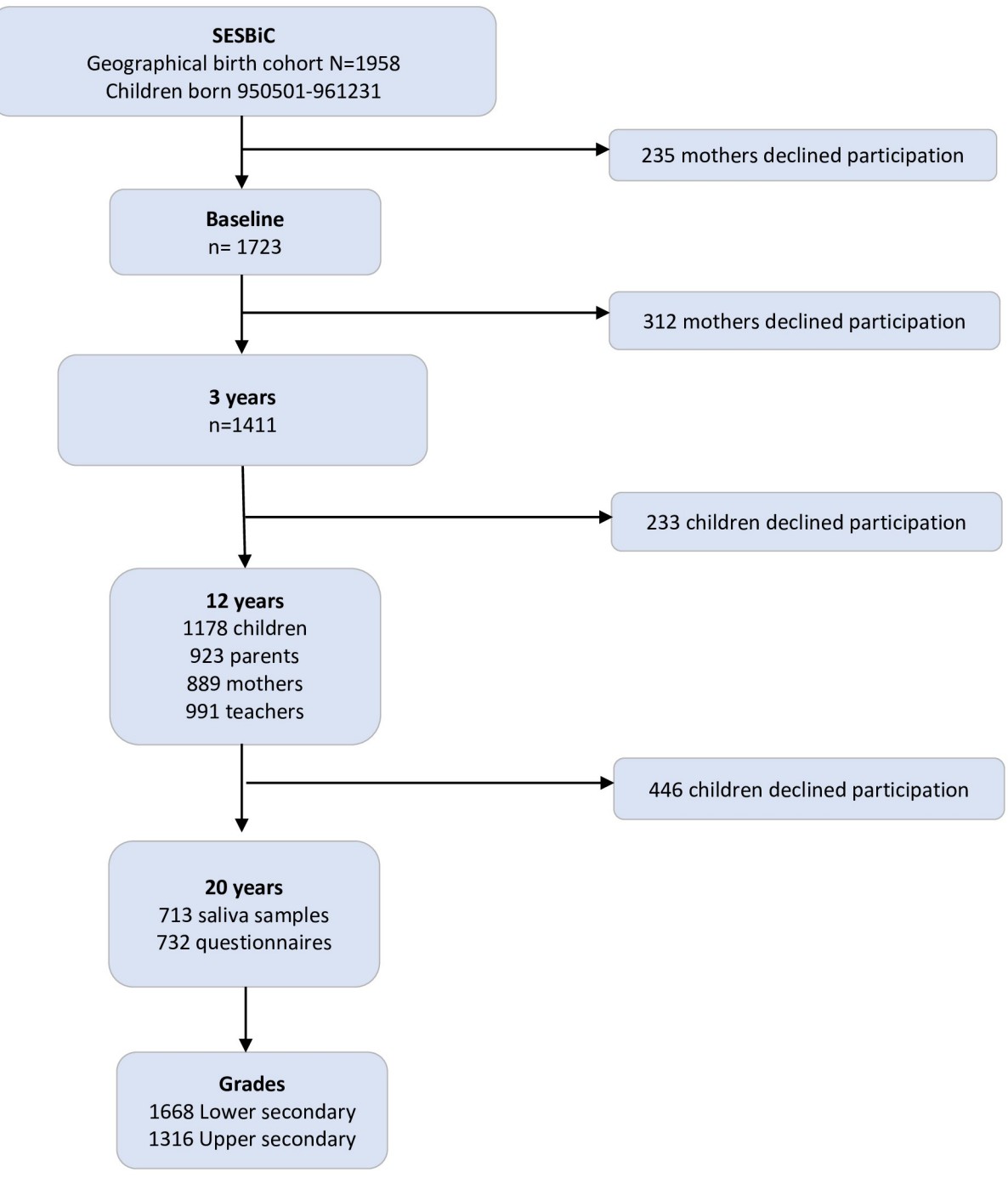

**Fig 2. Participation at each follow-up.**

the parents accepted participation they were instructed to sign and return the informed consent document enclosed in the letter.

*Register data.* Birth related information on the study participants was retrieved from the Swedish Medical Birth Register (MBR). Data retrieved included gestational age, birth weight, and size for gestational age and formed the basis for the variables preterm, low birthweight, SGA and LGA in this study.

*Grades.* In order to evaluate the cohort's academic performance at 15/16 and 18/19 years of age and the effect environmental, social, and behavioral factors have on their achievements, information on the individual's GPA was retrieved from national registers maintained by Statistics Sweden and the Swedish National Agency for Education. Information on the person's study results and whether the individual fulfilled the requirements to be accepted into each of the major programs were also retrieved.

## Definitions, questions and questionnaires

**Baseline.** Parental origin was divided into three levels "Both parents born in Sweden", "One immigrant parent", and "Two immigrant parents". Gender was defined as "male" or "female", preterm birth was defined as birth prior to 37 gestational weeks, and low birthweight was defined as a birthweight < 2500gr. Small for gestational age (SGA) and large for gestational age (LGA) were defined as either below 2SD of the gender and gestational age specific mean birthweight (SGA) or above 2SD of the gender and gestational age specific mean birthweight (LGA) [25].

*Edinburgh Postnatal Depression Scale (EPDS)* [26] was measured three months post-delivery. The EPDS was developed by Cox et al in 1987 and consist of ten questions. Each question is answered on a Likert scale, ranging between zero and three, with respect to depressive symptoms in the past seven days. The total score was dichotomized into <10, and > = 10, indicating the presence of depressive symptoms.

*Life stress score (LSS)* was measured at baseline, and in the current study the total score, as well as three sub-scales; stress due to medical factors (including areas of personal maturity, health status, workload, pregnancy related health, and consumption of health care), stress due to psychological factors (traumatic experiences/life events), and stress due to social factors (social network, education and employment, living arrangement) were used. The scores were dichotomized according to the 90th percentile, where those above the 90th percentile was considered to have experienced stress on each of the sub-scales.

**3-year follow-up.** From the 3-year follow-up information on maternal unemployment was dichotomized into "yes" and "no" as was parental divorce.

In this study a modified version of *Coddington's Life Event Scale (CLES)* was used, where only exposure and not the time aspect of the event was measured. CLES is used to screen for life events that may affect a person's development and growth [27, 28]. The number of life events was dichotomized into "scores > = 8" and "scores <8", according to the 90th percentile.

*Sense of Coherense (SOC)*, developed by Antonovsky in 1987, measures factors associated with a strong coping ability. The version used contains 13 questions, where every question was scored between one and seven, seven indicating a strong agreement [29–31]. The scale was used to measure maternal sense of coherence and dichotomized into "High" and "Low" based on the 90th percentile. The same dichotomization was used for the total score as well as the subscales internalizing, externalizing, destructive behavior, aggressive behaviour derived from *Child Behavior Check List (CBCL)* at three years of age. The CBCL comprises 113 questions measuring child behavior on eight different subscales, which can be aggregated into internalizing and externalizing problems [32]. Whereas child temperament at 3 years of age was dichotomized according to 75th percentile. The child's temperament was assessed using the *Difficult Child* questionnaire, developed by Tomas and Chess [33]. The questionnaire contains 11 questions where mothers were asked to assess their child's temperament in comparison to other children.

**12-year follow-up.** Living area was defined as "urban area" or "rural area" whereas living arrangement was categorized into "Living with mother and father", "Alternating between

mother and father" and "Other living arrangements" (including living with only one parent, foster parents and institutional care). Divorced parents was dichotomized into "Yes" and "No", maternal educational level when child was 12 years old was divided into "Primary, and Lower secondary", "Upper secondary", and "College/university". The child's self-esteem was measured using Rosenberg's self-esteem scale [34]. The scale comprises 10 questions which are answered on a 4-Likert scale where the answers ranges from "Strongly agree" to "Strongly disagree". The total score was divided into "< = 14" and "> = 15", as scores below 15 is considered to represent low self-esteem. In addition, CBCL/4-18 was used to estimate the sub-scales CBCL internalizing, CBCL externalizing as well as the CBCL total score were dichotomized into high and low according to the 90[th] percentile [35].

**Statistics.** Categorical data are presented as numbers (n) and percent (%), while continuous data are presented as mean and standard deviation (SD). Initially bivariate analyses including Pearson's Chi-square or Fisher's exact test (when cell count was below five) for evaluation of eligibility to secondary schooling and eligibility for college/university studies across socio-demographic background factors while mean GPA was evaluated using ANOVA.

Multiple logistic regression analysis with respect eligibility at each follow-up were modelled separately, where eligibility to upper secondary schooling and eligibility for college/university were defined as outcomes. To reduce the number of parameters being included in the model covering the entire study period, forward stepwise multiple regression models were performed for each time of follow-up. These stepwise multiple regression models, each outcome modelled separately, only included factors in Tables 1–3 that were found to be statistically significant. Statistical significance was defined as p<0.05 (two-sided). Cases with missing values were excluded from the analyses.

All analyses were performed using IBM SPSS, version 26 (IBM Inc. Armonk, NY, USA).

**Ethics.** The study has been approved by the Ethics Committee at the University of Lund in 1994 (LU439-93) and 1998 (LU124-98) and by the Regional Ethical Review Board at Linköping University in 2007 (M51-07), 2015 (M131-31).

## Results

### Eligibility to upper secondary education and university studies

Socio-demographic factors in relation to study performance is presented in Table 1. Parental origin, being born preterm or with low birthweight did not affect the eligibility to either upper secondary schooling or university studies. However, individuals born SGA were over-represented among those who did not fulfill the requirements for upper secondary schooling (4.1% vs 1.7%, p = 0.032) as well as university studies (2.5% vs 1.3%, p = 0.012). Moreover, being born LGA decreased the chances of fulfilling the requirements for being eligible for university studies (6.1% vs 3.5%, p = 0.021). Furthermore, individuals whose mothers scored >10 on the EPDS indicated symptoms of depression, were more likely of not being eligible for upper secondary school (16.7% vs 11.2%, p = 0.032) nor university studies (12.3% vs 10.5%, p = 0.020), Table 1. Maternal smoking was more than twice as common among children who did not fulfill the requirement for upper secondary school or college/university compared to children whose mothers did not smoke.

Maternal stressors at baseline were negatively associated with the children's study performance. Total score on the instrument LSS as well as the subscales "psychological" and "social stress" reduced the eligibility both with respect to upper secondary schooling (20.2% vs 6.9%, 10.9% vs 5.3%, and 14.2% vs 5.6%, respectively) and eligibility for university studies (11.4% vs 5.3%, 7.4% vs 4.1%, and 7.1% vs 4.2%, respectively), Table 1. Furthermore, the LSS subscale "medical stress" was negatively associated with school performance in lower secondary school

**Table 1. Baseline information, sociodemographic background, life stress scale factors, and birth characteristics.**

| | Eligibility to upper secondary schooling* | | | Overall** | | Eligibility to university studies* | | | | Overall*** | |
|---|---|---|---|---|---|---|---|---|---|---|---|
| | Not eligible N = 183 | Eligible N = 1485 | | GPA Lower Secondary School | | Did not start or complete upper secondary school N = 407 | Not eligible N = 297 | Eligible N = 1019 | | GPA Upper Secondary School | |
| | n (%) | n (%) | p- value | Mean/SD | p- value | n (%) | n (%) | n (%) | p-value | Mean/ SD | p-value |
| **Origin** | | | 0.425 | | 0.274 | | | | 0.082 | | 0.001 |
| Both parents born in Sweden | 148 (82.7) | 1256 (86.2) | | 206.47/61.77 | | 324 (81.6) | 260 (88.7) | 859 (86.1) | | 13.60/ 3.74 | |
| One immigrant parent | 15 (8.4) | 92 (6.3) | | 197.62/69.67 | | 36 (9.1) | 16 (5.5) | 61 (6.1) | | 13.35/ 4.28 | |
| Two immigrant parents | 16 (8.9) | 109 (7.5) | | 201.60/52.78 | | 37 (9.3) | 17 (5.8) | 78 (7.8) | | 12.19/ 4.46 | |
| **Gender** | | | 0.889 | | <0.001 | | | | <0.001 | | <0.001 |
| Male | 97 (53.0) | 779 (52.5) | | 197.74/59.37 | | 250 (61.4) | 181 (60.9) | 478 (46.9) | | 12.85/ 3.80 | |
| Female | 86 (47.0) | 706 (47.5) | | 216.91/62.58 | | 157 (38.6) | 116 (39.1) | 541 (53.1) | | 14.12/ 3.80 | |
| **Preterm** | | | 0.544 | | 0.892 | | | | 0.063 | | 0.040 |
| No | 174 (95.1) | 1424 (96.0) | | 205.28/61.56 | | 384 (97.2) | 278 (93.6) | 977 (95.9) | | 13.42/ 3.90 | |
| Yes | 9 (4.9) | 59 (4.0) | | 206.32/68.00 | | 11 (2.8) | 19 (6.4) | 42 (4.1) | | 14.43/ 2.44 | |
| **Low birthweight** | | | 0.811$ | | 0.775 | | | | 0.107 | | 0.268 |
| No | 147 (97.4) | 1184 (96.6) | | 205.70/62.18 | | 308 (97.8) | 241 (94.5) | 816 (96.6) | | 13.49/ 3.86 | |
| Yes | 4 (2.6) | 42 (3.4) | | 208.37/66.10 | | 7 (2.2) | 14 (5.5) | 29 (2.4) | | 14.14/ 2.53 | |
| **SGA** | | | 0.032 | | 0.082 | | | | 0.012 | | 0.021 |
| No | 164 (95.9) | 1397 (98.3) | | 205.65/61.69 | | 372 (96.1) | 271 (97.5) | 960 (98.7) | | 13.48/ 3.85 | |
| Yes | 7 (4.1) | 24 (1.7) | | 186.13/68.24 | | 15 (3.9) | 7 (2.5) | 13 (1.3) | | 11.70/ 5.24 | |
| **LGA** | | | 0.921 | | 0.620 | | | | 0.021 | | 0.223 |
| No | 165 (96.5) | 1369 (96.3) | | 205.12/61.79 | | 379 (97.9) | 261 (93.9) | 939 (96.5) | | 13.42/ 3.90 | |
| Yes | 6 (3.5) | 52 (3.7) | | 209.22/64.24 | | 8 (2.1) | 17 (6.1) | 34 (3.5) | | 14.08/ 3.33 | |
| **EPDS, above 10** | | | 0.032 | | 0.070 | | | | 0.020 | | 0.299 |
| No | 150 (83.3) | 1307 (88.8) | | 206.62/61.08 | | 329 (84.1) | 256 (87.7) | 908 (89.5) | | 13.51/ 3.83 | |
| Yes | 30 (16.7) | 165 (11.2) | | 198.12/65.49 | | 62 (15.9) | 36 (12.3) | 106 (10.5) | | 13.18/ 4.01 | |
| **Maternal smoking during pregnancy** | | | <0.001 | | <0.001 | | | | <0.001 | | <0.001 |
| No | 114 (62.3) | 1223 (82.4) | | 211.96/59.16 | | 287 (705) | 227 (76.4) | 867 (87.8) | | 13.71/ 3.73 | |
| Yes | 69 (37.7) | 262 (17.6) | | 178.23/65.35 | | 120 (29.5) | 70 (23.6) | 120 (12.2) | | 12.34/ 4.18 | |
| **Life Stress Score (LSS)** | | | | | | | | | <0.001 | | 0.006 |
| Total* | | | <0.001 | | <0.001 | | | | | | |

(*Continued*)

**Table 1.** (Continued)

| | Eligibility to upper secondary schooling* | | | Overall** | | Eligibility to university studies* | | | | Overall*** | |
|---|---|---|---|---|---|---|---|---|---|---|---|
| | Not eligible N = 183 | Eligible N = 1485 | | GPA Lower Secondary School | | Did not start or complete upper secondary school N = 407 | Not eligible N = 297 | Eligible N = 1019 | | GPA Upper Secondary School | |
| | n (%) | n (%) | p- value | Mean/SD | p- value | n (%) | n (%) | n (%) | p-value | Mean/SD | p-value |
| No (0–10) | 146 (79.8) | 1381 (93.1) | | 209.18/59.43 | | 339 (85.8) | 263 (88.6) | 965 (94.7) | | 13.54/3.84 | |
| Yes (11–23) | 37 (20.2) | 102 (6.9) | | 162.95/71.23 | | 56 (14.2) | 34 (11.4) | 54 (5.3) | | 12.45/3.85 | |
| **Medical**$^{\$\$}$ | | | **0.008** | | **0.005** | | | | **0.223** | | **0.424** |
| No (0–4) | 165 (90.2) | 1408 (94.9) | | 206.35/60.93 | | 367 (92.9) | 279 (93.9) | 970 (95.2) | | 13.45/3.86 | |
| Yes (5–8) | 18 (9.8) | 75 (5.1) | | 187.96/73.56 | | 28 (7.1) | 18 (6.1) | 49 (4.8) | | 13.82/3.78 | |
| **Psychological**$^{\$\$}$ | | | **0.002** | | **<0.001** | | | | **<0.001** | | **<0.001** |
| No (0–5) | 163 (89.1) | 1404 (94.7) | | 207.29/61.19 | | 225 (89.9) | 275 (92.6) | 977 (95.9) | | 13,56/3.81 | |
| Yes (6–10) | 20 (10.9) | 79 (5.3) | | 174.27/63.68 | | 40 (10.1) | 22 (7.4) | 42 (4.1) | | 11.79/4.26 | |
| **Social**$^{\$\$}$ | | | **<0.001** | | **<0.001** | | | | **<0.001** | | **<0.001** |
| No (0–4) | 157 (85.8) | 1400 (94.4) | | 208.09/60.27 | | 347 (87.8) | 276 (92.9) | 976 (95.8) | | 13.57/3.78 | |
| Yes (5–9) | 26 (14.2) | 83 (5.6) | | 165.83/69.94 | | 48 (12.2) | 21 (7.1) | 43 (4.2) | | 11.58/4.51 | |

*Numbers may not sum to population total due to missing data,

**UNIANOVA

$^{\$}$, Fisher's exact test,

$^{\$\$}$ above 90$^{th}$ percentile

and thus the eligibility to upper secondary school. However, medical factors were not associated with school performance in upper secondary school nor eligibility to college/university.

Table 2 shows that maternal unemployment at 3 years of age and having divorced parents at 3 years of age were negatively associated with being eligible for upper secondary schooling compared to an employed mother (17.9% vs 9.0%) and married parents (16.1% vs 9.1%). Similarly, having more than eight life events was more common among children who did not fulfill the requirement for upper secondary schooling (16.7% vs 6.5%). Children who scored high on the CBCL scales total score, and on the subscales externalizing, destructive behavior, and aggressive behaviour at age 3 were overrepresented among those who were not eligible for upper secondary schooling. Similar findings were detected on the eligibility for college/university education. Children with an unemployed mother, divorced parents, with more than eight life events, and whose mothers scored a low sense of coherence were more likely of not being eligible for college/university studies. In addition, scoring high on any of the CBCL scales (total score, externalizing problems, internalizing problems, destructive behaviour, and aggressive behavior) were all associated with an increased likelihood of ineligibility for college/university, Table 2.

From the 12-year follow-up alternating living or living with only one parent/foster parents were more frequent among children who did not fullfill the eligibility for secondary education

**Table 2.  Factors at the 3-year follow-up.**

| | Eligibility to upper secondary schooling* | | | Overall** | | Eligibility to university studies* | | | | Overall** | |
|---|---|---|---|---|---|---|---|---|---|---|---|
| | Not eligible N = 183 | Eligible N = 1485 | | GPA Lower Secondary School | | Did not start or complete upper secondary school N = 407 | Not eligible N = 297 | Eligible N = 1019 | | GPA Upper Secondary School | |
| | n (%) | n (%) | p-value | Mean/SD | p-value | n (%) | n (%) | n (%) | p-value | Mean/SD | p-value |
| **Unemployed mother** | | | **0.001** | | **<0.001** | | | | **<0.001** | | **<000.1** |
| No | 119 (82.1) | 1120 (91.0) | | 209.14/ 60.10 | | 262 (82.4) | 226 (91.5) | 784 (91.9) | | 13.66/ 3.71 | |
| Yes | 26 (17.9) | 111 (9.0) | | 181.64/ 72.05 | | 56 (17.6) | 21 (8.5) | 69 (8.1) | | 12.08/ 4.61 | |
| **Divorced parents** | | | **0.007** | | **<0.001** | | | | **<0.001** | | **<0.001** |
| Yes | 24 (16.1) | 113 (9.1) | | 175.80/ 71.90 | | 50 (17.0) | 37 (14.5) | 54 (6.3) | | 11.96/ 4.31 | |
| No | 125 (83.9) | 1125 (90.9) | | 209.36/ 59.77 | | 244 (83.0) | 218 (85.5) | 801 (93.7) | | 13.65/ 3.75 | |
| **Life event** | | | **<0.001** | | **<0.001** | | | | **0.017** | | **0.231** |
| At or below 8 | 125 (83.3) | 1158 (93.5) | | 208.09/ 60.28 | | 280 (88.9) | 233 (91.4) | 803 (93.8) | | 13.53/ 3.82 | |
| Above 8 | 25 (16.7) | 81 (6.5) | | 180.28/ 75.09 | | 35 (11.1) | 22 (8.6) | 53 (6.2) | | 13.02/ 4.02 | |
| **SoC, mother** | | | **0.389** | | **0.010** | | | | **0.011** | | **0.007** |
| No | 134 (91.2) | 1095 (88.8) | | 204.77/ 62.67 | | 270 (92.8) | 233 (91.7) | 743 (87.2) | | 13.42/ 3.88 | |
| Yes | 13 (8.8) | 138 (11.2) | | 218.51/ 54.49 | | 21 (7.2) | 21 (8.3) | 109 (12.3) | | 14.35/ 3.04 | |
| **SoC, above 75[th] percentile** | | | **0.523** | | **0.028** | | | | **0.224** | | **0.084** |
| No | 116 (78.9) | 944 (76.6) | | 204.26/ 62.94 | | 228 (78.4) | 204 (80.3) | 643 (75.5) | | 13.42/ 3.89 | |
| Yes | 31 (21.1) | 289 (23.4) | | 212.95/ 58.18 | | 63 (21.6) | 50 (19.7) | 209 (24.5) | | 13.86/ 3.49 | |
| **Temperament, Resilience** | | | **0.621** | | **0.213** | | | | **0.739** | | **0.803** |
| > = 17 (75[th] percentile) | 108 (72.0) | 911 (73.9) | | 207.32/ 62.06 | | 213 (72.7) | 192. (75.6) | 629 (73.4) | | 13.55/ 3.86 | |
| < = 16 | 42 (28.0) | 322 (26.1) | | 202.60/ 62.37 | | 80 (27.3) | 62 (24.4) | 224 (26.3) | | 13.49/ 3.58 | |
| **CBCL, total$** | | | **0.001** | | **<0.001** | | | | **0.002** | | **0.073** |
| No | 126 (83.4) | 1139 (91.9) | | 207.70/ 61.16 | | 252 (85.7) | 236 (91.8) | 791 (92.5) | | 13.58/ 3.81 | |
| Yes | 25 (16.6) | 100 (8.1) | | 186.88/ 68.65 | | 42 (14.3) | 21 (8.2) | 64 (7.5) | | 12.88/ 3.88 | |
| **Externalizing$** | | | **<0.001** | | **<0.001** | | | | **<0.001** | | **0.035** |
| No | 125 (82.8) | 1137 (91.8) | | 212.04/ 61.25 | | 269 (85.1) | 227 (88.3) | 798 (93.3) | | 13.59/ 3.78 | |
| Yes | 26 (17.2) | 102 (8.2) | | 200.31/ 62.42 | | 47 (14.9) | 30 (11.7) | 57 (6.7) | | 12-78/ 4.13 | |
| **Internalizing$** | | | **0.171** | | **0.007** | | | | **0.008** | | **0.156** |
| No | 133 (88.1) | 1133 (91.4) | | 208.07/ 61.84 | | 273 (86.4) | 230 (89.5) | 789 (92.3) | | 13.57/ 3.83 | |
| Yes | 18 (11.9) | 106 (8.6) | | 196.99/ 62.56 | | 43 (13.6) | 27 (10.5) | 66 (7.7) | | 13.02/ 3.63 | |

*(Continued)*

**Table 2.** (Continued)

| | Eligibility to upper secondary schooling* | | | Overall** | | Eligibility to university studies* | | | | Overall** | |
|---|---|---|---|---|---|---|---|---|---|---|---|
| | Not eligible N = 183 | Eligible N = 1485 | | GPA Lower Secondary School | | Did not start or complete upper secondary school N = 407 | Not eligible N = 297 | Eligible N = 1019 | | GPA Upper Secondary School | |
| | n (%) | n (%) | p-value | Mean/ SD | p-value | n (%) | n (%) | n (%) | p-value | Mean/ SD | p-value |
| **Destructive$** | | | 0.002 | | <0.001 | | | | <0.001 | | 0.039 |
| No | 129 (85.4) | 1150 (92.8) | | 207.79/ 61.25 | | 254 (86.4) | 230 (89.5) | 808 (94.5) | | 13.58/ 3.80 | |
| Yes | 22 (14.6) | 89 (7.2) | | 183.20/ 67.77 | | 40 (13.6) | 27 (10.5) | 47 (5.5) | | 12.72/ 3.93 | |
| **Aggressive$** | | | 0.005 | | 0.001 | | | | 0.009 | | 0.204 |
| No | 130 (86.1) | 1148 (92.7) | | 207.44/ 61.24 | | 259 (88.1) | 233 (90.7) | 800 (93.6) | | 13.56/ 3.80 | |
| Yes | 21 (13.9) | 91 (7.3) | | 187.41/ 69.13 | | 35 (11.9) | 24 (9.3) | 55 (6.4) | | 13.04/ 4.03 | |

*Numbers may not sum to population total due to missing data,

**UNIANOVA

$, above 90th percentile

as were low maternal educational level, Table 3. In addition, children with a low self-esteem or with high scores on the total score on CBCL or its subscale externalizing problems were more common among those who did not fullfil the requirements for secondary education. Regarding eligibility for college/university studies, children living in rural areas, children who did not live with both their mother and father, whose parents were divorced, whose mother's had a low educational level were all associated with children not being eligible for college/university studies. Similarly, scoring high on all CBCL scales (total score, internalizing problems, and externalizing problems) was more common among children who were ineligible for college/ university studies, Table 3.

## Grade point average from lower and upper secondary schooling

In general, females finished their education with higher grades compared to males both in lower secondary school as well as upper secondary school (216.91 vs 197.74 and 14.12 vs 12.85, respectively both p<0.001), while maternal smoking was negatively associated with their GPA in both lower and upper secondary school compared to children whose mothers did not smoke, Table 1. Being born with non-optimal birth characteristics did not have an effect on GPA from lower secondary school. However, being born preterm and SGA was associated with a lower GPA from upper secondary school (13.42 vs 14.43 and 11.70 vs 13.48, respectively). As for eligibility to the next level of education, maternal stressors at baseline were negatively associated with the children's GPA at both lower and upper secondary school. Maternal experience on all four scales of stressors (total, medical, psychological and social) was associated with a decreased the GPA at lower secondary school. Similar findings were present for GPA at upper secondary school, apart from the medical subscale were no association was found, Table 1.

Having an unemployed mother or divorced parents at three years of age were associated with a lower GPA at both lower and upper secondary school while eight or more life events at three years of age only affected the GPA at lower secondary school, Table 2. In addition,

**Table 3. Factors at the 12-year follow-up.**

| | Eligibility to upper secondary schooling* | | | Overall** | | Eligibility to university studies* | | | | Overall** | |
|---|---|---|---|---|---|---|---|---|---|---|---|
| | Not eligible N = 183 | Eligible N = 1485 | | GPA Lower Secondary School | | Did not start or complete upper secondary school N = 407 | Not eligible N = 297 | Eligible N = 1019 | | GPA Upper Secondary School | |
| | n (%) | n (%) | p- | Mean/SD | p | n (%) | n (%) | n (%) | p-value | Mean/SD | p-value |
| **Mother's alcohol habits** | | | 0.722* | | 0.004 | | | | 0.016 | | 0.001 |
| Low consumer | 56 (98.2) | 786 (95.9) | | 214.21/56.7 | | 140 (97.2) | 166 (99.4) | 545 (94.6) | | 13.73/3.68 | |
| High consumer, 2–4 times per week | 1 (1.8) | 34 (4.1) | | 242.71/52.30 | | 4 (2.8) | 1 (0.6) | 31 (5.4) | | 15.91/2.16 | |
| **Living—municipality** | | | 0.795 | | 0.861 | | | | <0.001 | | 0.305 |
| Urban | 40 (71.4) | 568 (69.8) | | 215.55/58.78 | | 110 (78.0) | 96 (57.5) | 410 (71.7) | | 13.75/3.88 | |
| Rural | 16 (28.6) | 246 (30.2) | | 214.81/52.24 | | 31 (22.0) | 71 (42.5) | 162 (28.3) | | 14.03/2.94 | |
| **Living arrangement** | | | <0.001 | | <0.001 | | | | <0.001 | | <0.001 |
| Mother *and* Father | 31 (56.4) | 599 (78.3) | | 220.67/54.35 | | 89 (65.4) | 111 (69.8) | 436 (81.5) | | 14.19/3.37 | |
| Alternating, mother and father | 9 (16.4) | 78 (10.2) | | 196.90/51.72 | | 14 (10.3) | 21 (13.2) | 52 (9.7) | | 12.86/3.53 | |
| Other living arrangements | 15 (27.3) | 88 (11.5) | | 190.80/66.24 | | 33 (24.3) | 27 (17.0) | 47 (8.8) | | 12.12/4.53 | |
| **Divorce** | | | 0.335 | | <0.001 | | | | 0.008 | | <0.001 |
| No | 86 (80.4) | 892 (84.0) | | 213.23/58.66 | | 182 (79.8) | 168 (78.5) | 637 (86.1) | | 13.80/3.75 | |
| Yes | 21 (19.6) | 170 (16.0) | | 196.35/58.20 | | 46 (20.2) | 46 (21.5) | 103 (13.9) | | 12.63/4.03 | |
| **Educational level mother** | | | 0.003 | | <0.001 | | | | <0.001 | | <0.001 |
| College/university | 12 (21.4) | 331 (40.4) | | 230.09/56.29 | | 46 (32.2) | 44 (26.3) | 257 (44.7) | | 14.30/3.73 | |
| Primary and Lower secondary school | 35 (62.5) | 430 (52.5) | | 209.24/53.99 | | 73 (51.0) | 104 (62.3) | 290 (50.4) | | 13.67/3.43 | |
| Upper secondary school | 9 (16.1) | 58 (7.1) | | 183.66/57.98 | | 24 (16.8) | 19 (11.4) | 28 (4.9) | | 12.09/4.37 | |
| **Self-esteem** | | | 0.007 | | <0.001 | | | | 0.100 | | 0.021 |
| < = 14 | 41 (38.7) | 275 (26.3) | | 196.50/60.99 | | 69 (32.7) | 61 (29.0) | 186(25.5) | | 13.16/4.02 | |
| > = 15 | 65 (61.3) | 770(73.7) | | 215.82/57.49 | | 142 (67.3) | 149 (71.0) | 544 (74.5) | | 13.77/3.75 | |
| **CBCL, total$** | | | <0.001 | | <0.001 | | | | <0.001 | | <0.001 |
| No | 46 (78.0) | 787 (91.7) | | 219.01/53.98 | | 116 (79.5) | 148 (85.5) | 574 (95.0) | | 13.98/3.59 | |
| Yes | 13 (22.0) | 71 (8.3) | | 173.63/67.02 | | 30 (20.5) | 25 (14.5) | 30 (5.0) | | 11.98/3.73 | |
| **Internalizing$** | | | 0.092 | | 0.002 | | | | 0.003 | | 0.061 |
| No | 51 (86.4) | 794 (92.5) | | 216.52/55.62 | | 126 (86.3) | 156 (90.2) | 569 (94.2) | | 13.88/3.67 | |
| Yes | 8 (13.6) | 64 (7.5) | | 195.21/69.92 | | 20 (13.7) | 17 (9.8) | 35 (5.8) | | 12.99/3.20 | |
| **Externalizing$** | | | 0.001 | | <0.001 | | | | <0.001 | | <0.001 |

*(Continued)*

**Table 3.** (Continued)

| | Eligibility to upper secondary schooling[*] | | | Overall[**] | | Eligibility to university studies[*] | | | | Overall[**] | |
|---|---|---|---|---|---|---|---|---|---|---|---|
| | Not eligible N = 183 | Eligible N = 1485 | | GPA Lower Secondary School | | Did not start or complete upper secondary school N = 407 | Not eligible N = 297 | Eligible N = 1019 | | GPA Upper Secondary School | |
| | n (%) | n (%) | p- | Mean/ SD | p | n (%) | n (%) | n (%) | p-value | Mean/ SD | p-value |
| No | 46 (78.0) | 784 (91.4) | | 218.97/ 55.05 | | 117 (80.1) | 146 (84.4) | 571 (94.5) | | 13.99/ 3.60 | |
| Yes | 13 (22.0) | 74 (8.6) | | 175.57/ 58.48 | | 29 (19.9) | 27 (15.6) | 33 (5.5) | | 12.06/ 3.58 | |

[*]Numbers may not sum to population total due to missing data,

[**]UNIANOVA

$, above 90[th] percentile

externalizing problems and destructive behavior at three years of age were more common among those with a lower GPA at both lower and upper secondary school, while internalizing problems, aggressive behavior and the CBCL total score also were overrepresented among those with a lower GPA from lower secondary school, Table 2.

Factors at 12 years of age found to negatively affect GPA at both lower and upper secondary school included: not living with mother and/or father, divorced parents, low level of education among mothers, and having a low self-esteem at 12 years of age. Moreover, externalizing problems and total CBCL problems at age 12 were associated with a lower GPA at both lower and upper secondary school, Table 3.

**Multiple logistic regression models.** To evaluate what predicts future failure in lower and upper secondary schooling multiple logistic regressions were performed within each follow-up. These models were restricted to, given the large number of predictors, stepwise regression models only including factors that were statistically significant in the bivariate analyses presented in Tables 1–3, each table modelled separately.

In the analyses of eligibility to upper secondary school it was found that children who were born SGA and children whose mothers had experienced post-partum depression symptoms on EPDS were less likely to be eligible for secondary schooling (OR = 0.61, 95% CI = 0.39–0.95 and OR = 0.39, 95% CI = 0.17–0.93, respectively) compared to children born AGA or children whose mothers scored low on EPDS. Including the total score on LSS and maternal smoking altered these findings and neither SGA nor high scores on EPDS were associated with school failure. Instead, the total score of LSS and maternal smoking were the only remaining variables. Children to mothers who had experienced life stressors at baseline were less likely to be eligible for upper secondary schooling compared to children where the mothers exhibited low level of stress on the total score (OR = 0.32, 95% CI = 0.21–0.49, Table 4) as were children to mothers who smoked (OR = 0.44, 95% CI = 0.31–0.63).

Factors from the three-year follow-up predicting non-eligibility to upper secondary school included unemployed mother (OR = 0.58, 95% CI = 0.35–0.98), more than eight life events (OR = 0.42, 95% CI = 0.25–0.72), and having more externalizing problems (OR = 0.51, 95% CI = 0.31–0.85), Table 4.

Having alternating living arrangements (OR = 0.46, 95% CI = 0.21–1.02) or having other living arrangements (e.g., foster parents or institutional care) at 12-years of age (OR = 0.33, 95% CI = 0.17–0.65) compared to living with both parents, and scoring above the 90[th] percentile on the CBCL total score (OR = 0.35, 95% CI = 0.17–0.71) decreased the

**Table 4. Multiple logistic regression analyses on the eligibility to upper secondary school and college/university.**

| | Eligibility to upper secondary schooling | | Eligibility to university studies | |
|---|---|---|---|---|
| | OR (95% CI)* | OR (95% CI)** | OR (95% CI)* | OR (95% CI)** |
| Table 1. **Baseline** | | | | |
| **Gender** | | | | |
| Boy | *Not included* | | 0.57 (0.43–0.75) | 0.57 (0.38–0.85) |
| Girls | *Not included* | | *Reference* | *Reference* |
| **LGA** | | | | |
| No | *Not included* | | *Reference* | |
| Yes | *Not included* | | 0.53 (0.28–0.98) | |
| **Maternal smoking** | | | | |
| No | *Reference* | | *Reference* | |
| Yes | 0.44 (0.31–0.63) | | 0.65 (0.46–0.91) | |
| **Bas_LSS_totcut90** | | | | |
| No | *Reference* | | *Reference* | *Reference* |
| Yes | 0.32 (0.21–0.49) | | 0.47 (0.29–0.76) | 0.32 (0.14–0.77) |
| Table 2. **3 year** | | | | |
| **Unemployed mother** | | | | |
| No | *Reference* | | *Not included* | |
| Yes | 0.58 (0.35–0.98) | | *Not included* | |
| **Divorce** | | | | |
| No | *Not included* | | *Reference* | *Reference* |
| Yes | *Not included* | | 0.40 (0.26–0.64) | 0.30 (0.16–0.58) |
| **Life event** | | | | |
| No | *Reference* | | *Not included* | |
| Yes | 0.42 (0.25–0.72) | | *Not included* | |
| **CBCL destructive** | | | | |
| No | *Not included* | | *Reference* | |
| Yes | *Not included* | | 0.56 (0.33–0.95) | |
| **Externalizing** | | | | |
| No | *Reference* | | *Not included* | |
| Yes | 0.51 (0.31–0.85) | | *Not included* | |
| Table 3. **12 year** | | | | |
| **Living—municipality** | | | | |
| Urban | *Not included* | | *Reference* | *Reference* |
| Rural | *Not included* | | 0.53 (0.36–0.78) | 0.48 (0.32–0.72) |
| **Divorce parents** | *Not included* | | | |
| No | *Not included* | | *Reference* | |
| Yes | *Not included* | | 0.58 (0.37–0.92) | |
| **Living arrangements** | | | | |
| Mother and father | *Reference* | *Reference* | *Not included* | |
| Alternating, mother and father | 0.46 (0.21–1.02) | 0.46 (0.21–1.02) | *Not included* | |
| Other arrangements | 0.33 (0.17–0.65) | 0.33 (0.17–0.65) | *Not included* | |
| **Educational level mother** | | | | |
| College/university | *Not included* | | *Reference* | *Reference* |
| Primary and Lower secondary school | *Not included* | | 0.53 (0.35–0.81)) | 0.52 (0.34–0.81) |
| Upper secondary school | *Not included* | | 0.21 (0.10–0.44) | 0.28 (0.12–0.62) |
| **CBCL, total score** | | | | |
| No | *Reference* | *Reference* | *Not included* | |

*(Continued)*

**Table 4.** (Continued)

| | Eligibility to upper secondary schooling | | Eligibility to university studies | |
|---|---|---|---|---|
| | OR (95% CI)* | OR (95% CI)** | OR (95% CI)* | OR (95% CI)** |
| Yes | 0.35 (0.17–0.71) | 0.35 (0.17–0.71) | *Not included* | |
| **Externalizing** | | | | |
| No | *Not included* | | *Reference* | *Reference* |
| Yes | *Not included* | | 0.36 (0.20–0.67) | 0.45 (0.24–0.86) |

*Stepwise logistic regression models performed per table, separate models for each outcome. Each model only included factors from Tables 1–3 that were statistically significant.

**Stepwise logistic regression model including all statistically significant variables from the table-wise models.

likelihood of eligibility to upper secondary schooling. To further elucidate the which variables were the most associated with non-eligibility to upper secondary schooling stepwise logistic regression models including all variables simultaneously were performed. These models revealed that a high total score on CBCL at 12-years of age (OR = 0.35, 95 = % CI = 0.17–0.71) and having other living arrangements (e.g., foster parents or institutional care) at 12-years of age (OR = 0.33, 95% CI = 0.17–0.65) decreased the likelihood of fulfilling the requirements for upper secondary school.

In terms of eligibility to college/university boys were less likely to be eligible (OR = 0.57, 95% CI = 0.43–0.75) compared to girls. Also, maternal smoking (OR = 0.65, 95% CI = 0.46–0.91) and low maternal scores on the total life stress scale at baseline (OR = 0.47, 95% CI = 0.29–0.76), having divorced parents (OR = 0.40, 95% CI = 0.26–0.64) as well as exhibiting destructive behavior at three years of age (OR = 0.56, 95% CI = 0.33–0.95) all decreased the likelihood of being eligible for college/university, Table 4. Eligibility to college/university was negatively affected by rural living (OR = 0.53, 95% CI = 0.36–0.78) compared to urban living, having divorced parents (OR = 0.58, 95% CI = 0.37–0.92), having a mother with elementary (OR = 0.53, 95% CI = 0.35–0.81) or high school education (OR = 0.21, 95% CI = 0.10–0.44) and having exhibited externalizing problems at 12 years of age (OR = 0.36, 95% CI = 0.20–0.67). As for eligibility to upper secondary school, stepwise multiple logistic regression models were performed to identify variables most associated with eligibility to college/university. This analysis showed that male gender (OR = 0.57, 95% CI = 0.38–0.85) and maternal total stress at baseline (OR = 0.32, 95% CI = 0.14–0.77) were negatively associated with eligibility. Having divorced parents at three years of age (OR = 0.30, 95% CI = 0.16–0.58), rural living (OR = 0.48, 95% CI = 0.32–0.72), maternal elementary (OR = 0.52, 95% CI = (0.34–0.81) or maternal high school (OR = 0.28, 95% CI = 0.12–0.62) and exhibiting externalizing problems at 12 years of age (OR = 0.45, 95% CI = 0.24–0.91) were all associated with a decreased likelihood of being eligible for college/university education.

## Discussion

In this longitudinal study children's academic performance, measured as eligibility to upper secondary school and college/university, was found to be affected by socio-demographic and behavioral factors from birth up to 12 years of age. The results, arranged on the basis of previous research areas could be summarized in four findings.

First, boys in general, but especially concerning school performance (overall GPA) and eligibility to higher education did worse than girls. This is in concordance with the general gender difference in study performance in Sweden and was also present the year these adolescents

could be expected to graduate from lower and upper secondary school (https://www.skolverket.se/download/18.6bfaca41169863e6a65a470/1553964853546/pdf2979.pdf).

Second, bivariate analyses of birth related data showed that children born SGA were found to be less likely to fulfill the requirements for upper secondary schooling compared to those who were not born SGA. This may be explained by a reduced cognitive ability among children born SGA as shown both in a large study of SGA children in the US [5] as well as a study in Sweden using the Stockholm Birth Cohort [36]. Also, even though not verified in the currents study, previous studies have shown that children born with a low birthweight as well as children born SGA are more likely to experience cognitive problems and neurodevelopmental problems [37–39]. However, some of these differences may be reduced or diminish as the child grows.

Maternal smoking can be viewed as an expression for stress and as in several other studies, the current study found a negative association between scholastic performance and maternal smoking during pregnancy. Maternal smoking during pregnancy has been linked to adverse pregnancy and delivery outcomes but also with impaired cognitive and neurological development and thus potentially increasing the risk for poorer educational performance. However, it is possible that this association could be explained by other factors as smoking is more commen among women, e.g., with a low income or with a lower educational level.

Third, maternal life stress in the child's early years appears to be extremely important. Children to mothers experiencing high levels of stress were found to be less likely to be eligible for upper secondary school as well as college/university. A possible reason for this may be the exposure to stressors as early as in the intrauterine environment. This is in line with the findings in the current study as the LSS scales, which were measured during pregnancy, remain important factors both with respect to the children's eligibility to higher education but also with respect to the children's GPA. The LSS can be considered a proxy for maternal vulnerability at time of pregnancy and childbirth, where the elevated risk due to social factors highlights the importance of a functioning social network for the mother. Whereas the increased risk for children whose mothers indicated problems due to psychological factors, stresses the importance of maternal upbringing and the risk for heredity of "social belonging". Other contributing factors can be exposure to stress for a prolonged period of time, or several stressors occurring during a shorter timeperiod. This is partly supported in the current study as children who at time of entering high school had experienced early life events, parental divorce and reported either "alternating between mother and father" or other living arrangements were at an increased risk for both ineligibility to higher education as well as a lower GPA at both lower and upper secondary school.

Fourth, child behavior in relation to academic performance and drop out from school has been widely studied, the findings have been somewhat diverging. While most studies conclude that both internalizing and externalizing problems are associated with a lowered educational attainment, in the present study, it was found that externalizing problems at both 3 and 12 years of age were associated with an increased likelihood for ineligibility to upper secondary school and college/university as well as reduced GPA. A study on social selection and social causation effects in the present population found behavioral problems at age 3 to be n associated with performing below grade at age 12, and emotional and conduct problems at age 12 increased the risk for lack of complete final grades from compulsory school and non-eligibility to higher education [40].

In contrast to many previous studies, the negative association between internalizing behavior at 3 and 12 years of age and eligibility to secondary schooling was not found. However, internalizing behavior was found to affect GPA at lower secondary school as well as eligibility

to college/university. Thus, it appears as if internalizing problems do not affect the students until upper secondary school. It has been theorized that children with externalizing problems later on develop internalizing problems affecting their school performance negatively [41].

### Strengths

A major strength in this study is the longitudinal design with several follow-ups and inclusion of validated register data from Statistics Sweden and the National Board of Health and Welfare, enabling models including factors from birth up to the most recent follow-up. Using register data allows for inclusion of factors where there are no recollection bias. Another strength is the diversity of information included in the study–socio-economic data from baseline, birth characteristics, child behavior from 3 and 12 years of age, GPA from the final years in lower and upper secondary school, as well as eligibility to upper secondary school and college/university.

### Limitations

Unfortunately, there are some concerns regarding the participation rate in the current study. Since the start of the study in 1995/1996 the participation rate has decreased with each follow-up. This is a quite common problem in longitudinal studies extending over a long period–in this case approximately 20 years. However, even though the non-response rate is rather high, drop-out analyses did not indicate significant difference with respect to maternal civil status. However, significant differences with respect to baseline factors such as gender of the child, parental immigrant background were found and may affect the findings. Another possible limitation is the potential risk for missing important confounders not available in the registers or as a question in one of the questionnaires. Finally, there was no information available if the children had received some support such as e.g. special education to improve their study results. The lack of this information means that we can't estimate the importance of interventions for some of the variables and that without interventions the differences concerning both eligibility and GPA could have been even more evident.

### Conclusion

Maternal stressors during the child's first years, and externalizing behaviour problems at three years of age and total problem scores on CBCL at 12 years of age appears to be associated with GPA from lower and upper secondary school. An association between these factors was also appears to be present regarding eligibility to both upper secondary school and college/university.

Therefore, order to provide children with improved conditions to complete at least upper secondary school, it is important to identify mothers with high stressors dduring pregnancy and during the child's first years in life to offer guidance and support to reduce the effects on the children's study performance.

### Author Contributions

**Conceptualization:** Marie Bladh.

**Formal analysis:** Marie Bladh.

**Methodology:** Sara Agnafors.

**Supervision:** Carl Göran Svedin.

**Validation:** Sara Agnafors.

**Writing – original draft:** Marie Bladh.

**Writing – review & editing:** Marie Bladh, Carl Göran Svedin, Sara Agnafors.

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
