## [Decision Letter · Decision Letter 0]

22 Dec 2021

PONE-D-21-22136Predictors of educational failure at 16 and 19 years of age – SESBiC longitudinal study.PLOS ONE

Dear Dr. Bladh,

Thank you for submitting your manuscript to PLOS ONE. After careful consideration, we feel that it has merit but does not fully meet PLOS ONE’s publication criteria as it currently stands. Therefore, we invite you to submit a revised version of the manuscript that addresses the points raised during the review process.

We look forward to receiving your revised manuscript.

Kind regards,

Yacob Zereyesus, Ph.D.

Academic Editor

PLOS ONE

Journal Requirements:

Reviewers' comments:

Reviewer's Responses to Questions

**Comments to the Author**

1. Is the manuscript technically sound, and do the data support the conclusions?

Reviewer #1: Partly

2. Has the statistical analysis been performed appropriately and rigorously? 

Reviewer #1: I Don't Know

3. Have the authors made all data underlying the findings in their manuscript fully available?

Reviewer #1: No

4. Is the manuscript presented in an intelligible fashion and written in standard English?

Reviewer #1: Yes

5. Review Comments to the Author

Reviewer #1: In this study the authors have followed a longitudinal cohort of children from birth to finishing secondary school and the explore different parental and individual exposures and their association with education attainment. Although there are some minor and unimportant typos, the language is clear and the presentation of the study is straightforward. I have, however some objections.

1) The tables are not completely explained. For example: in Table 3, the figures don’t add up when checking the columns. This is probably due to missing data, and since the proportion of missing data varies between different variables and, probably, categories, this should better be presented in a way that is easier to grasp.

2) There are also errors in the table 3: The number of not eligible children for the variable Self-esteem is reported to be 730 and there are no eligible children.

3) Interpretation of the data. The authors conclude that externalizing behavior among the children and maternal stress should be identified in order to improve the chances of successful education attainment. These factors, however, are present only in 20.2% and 17.2% of all the children that are not eligible for upper secondary school, respectively, whereas 37.7% of the children whose mothers were smoking during pregnancy are not eligible. Smoking during pregnancy should therefore, in my opinion, be a more important factor. The authors findings are interesting, but their conclusion is somewhat too far-reaching.

6. PLOS authors have the option to publish the peer review history of their article (what does this mean?). If published, this will include your full peer review and any attached files.

Reviewer #1: No

---

## [Author Response · Author response to Decision Letter 0]

2 Aug 2022

Dear Editor and Reviewer,

Thank you for identifying areas of improvement in this manuscript. 

Below follows answers and comments to the concerns mentioned by the reviewer under question 5 in the review process.

We have done our utmost to accommodate requested changes to the manuscript, and hopefully you will find the changes adequate.

Looking forward to hearing back from you.

Best regards,

Marie Bladh

Reviewer #1: In this study the authors have followed a longitudinal cohort of children from birth to finishing secondary school and the explore different parental and individual exposures and their association with education attainment. Although there are some minor and unimportant typos, the language is clear and the presentation of the study is straightforward. 

I have, however some objections.

1) The tables are not completely explained. For example: in Table 3, the figures don’t add up when checking the columns. This is probably due to missing data, and since the proportion of missing data varies between different variables and, probably, categories, this should better be presented in a way that is easier to grasp.

Answer: We understand the confusion regarding the number included in the analyses. Therefore, the total numbers have been added for each column (in the head of each table), and a footnote has been added explaining that not all numbers sum up to the total due to missing data.

2) There are also errors in the table 3: The number of not eligible children for the variable Self-esteem is reported to be 730 and there are no eligible children.

Answer: Thank you for noticing this! You are correct regarding Table 3. Unfortunately, the numbers for Self-esteem had been shifted to the left removing the numbers that should have been presented in the first column. This has now been corrected.

3) Interpretation of the data. The authors conclude that externalizing behavior among the children and maternal stress should be identified in order to improve the chances of successful education attainment. These factors, however, are present only in 20.2% and 17.2% of all the children that are not eligible for upper secondary school, respectively, whereas 37.7% of the children whose mothers were smoking during pregnancy are not eligible. Smoking during pregnancy should therefore, in my opinion, be a more important factor. The authors findings are interesting, but their conclusion is somewhat too far-reaching.

Answer: We partly agree with the reviewer on this matter. Yes, a higher proportion of children non-eligible to upper secondary school or university/college had mothers who smoked during pregnancy. Also, as mentioned in the discussion it is possible that it is not smoking per se that is a risk factor but could be considered as a proxy variable for stress. 

The conclusion has been somewhat rephrased to include smoking and its relation to stress, while also nuancing the conclusion.

---

## [Decision Letter · Decision Letter 1]

24 Oct 2022

PONE-D-21-22136R1Predictors of educational failure at 16 and 19 years of age – SESBiC longitudinal study.PLOS ONE

Dear Dr. Bladh,

Thank you for submitting your manuscript to PLOS ONE. After careful consideration, we feel that it has merit but does not fully meet PLOS ONE’s publication criteria as it currently stands. Therefore, we invite you to submit a revised version of the manuscript that addresses the points raised during the review process. Reviewer has suggested a few points to improve the paper and I would appreciate if you could address and include those concerns in the revised mansucript by Nov 4, 2022. 

We look forward to receiving your revised manuscript.

Kind regards,

Santosh Kumar

Academic Editor

PLOS ONE

Reviewers' comments:

Reviewer's Responses to Questions

**Comments to the Author**

1. If the authors have adequately addressed your comments raised in a previous round of review and you feel that this manuscript is now acceptable for publication, you may indicate that here to bypass the “Comments to the Author” section, enter your conflict of interest statement in the “Confidential to Editor” section, and submit your "Accept" recommendation.

Reviewer #1: (No Response)

2. Is the manuscript technically sound, and do the data support the conclusions?

Reviewer #1: Partly

3. Has the statistical analysis been performed appropriately and rigorously? 

Reviewer #1: Yes

4. Have the authors made all data underlying the findings in their manuscript fully available?

Reviewer #1: (No Response)

5. Is the manuscript presented in an intelligible fashion and written in standard English?

Reviewer #1: Yes

6. Review Comments to the Author

Reviewer #1: Predictors of educational failure at 16 and 19 years of age – SESBiC longitudinal study.

In this study the authors present analyses from a longitudinal study on children born in the mid-1990s in southern Sweden and factors associated with school grades. The study is interesting and apart from some minor typing errors, the it is well written and easy to comprehend. I have, however, some comments.

Major comments

1. There is a huge difference between “significant” and “statistically significant”. In the conclusion of the abstract, the authors claim that “Identifying mothers smoking during pregnancy or with high level of stressors as well as children with externalizing behaviour problems to provide guidance and support is very important as these two factors appear to be associated with future study performance in both lower and upper secondary school”. Although the results are interesting, they do not support such a claim. For example, only 69 of 339 children exposed to maternal smoking in utero were ineligible for upper secondary school. The positive predictive value was thus only 0.21. Since there are no tests in the study of the precision with which educational failure can be predicted on the basis of the covariates under study, I think the authors should abstain from drawing clinical conclusions that are not supported by their results.

2. In the Discussion, it is suggested that the association between maternal smoking during pregnancy and offspring’s school performance is a casual one. This may definitely be so, but given that maternal IQ and smoking during pregnancy also have been shown to correlate, the association found by the authors could have other explanations. Since correlation does not prove causation, the authors should be more prudent in the interpretation of their data.

3. In Table 4, missing values appears to have huge effects on the analysis since 7 out of 18 analysis cannot be performed because of missing data. How many study subjects are included in the final models? Given the appearance of the table, I would assume this number to be rather low. Have the authors tried to address this problem?

Minor comments:

1. In the abstract, the abbreviation “CBCL” is not explained. For those not in the field, this makes the abstract less comprehensible.

2. Is it really necessary to introduce the abbreviation GPA?

3. In the methods, second paragraph, the sentence “….yet viewed as a an education one must complete” suggests, according to my interpretation, that upper secondary schooling is more a question of social and peer pressure than about, for example, increasing the individuals chances of finding a job in the future. Even though I agree with the authors to a large extent, I find this phrasing a little unidimensional.

4. In the Study Population section, “five municipalities in southern Sweden” is a little vague. Is there a thought behind this way of presenting the study population?

5. In the 3-year follow-up, the number of life events was dichotomized in <=8 and >8. Why this cut off?

6. At the end of the same paragraph, there are minor typos (“percentilen” should be translated into English, and “child” is misspelt)

7. In the 12-year follow-up, there are also cut offs for dichotomization that are not explained.

8. Table 4. Does the table show all variables included in the model? This is not entirely clear.

7. PLOS authors have the option to publish the peer review history of their article (what does this mean?). If published, this will include your full peer review and any attached files.

Reviewer #1: No

---

## [Author Response · Author response to Decision Letter 1]

24 Nov 2022

Dear reviewer, 

Below you will find the responses and descriptions on changes made according to suggestions. Hopefully you will find the changes satisfactory.

Major comments

1. There is a huge difference between “significant” and “statistically significant”. In the conclusion of the abstract, the authors claim that “Identifying mothers smoking during pregnancy or with high level of stressors as well as children with externalizing behaviour problems to provide guidance and support is very important as these two factors appear to be associated with future study performance in both lower and upper secondary school”. Although the results are interesting, they do not support such a claim. For example, only 69 of 339 children exposed to maternal smoking in utero were ineligible for upper secondary school. The positive predictive value was thus only 0.21. Since there are no tests in the study of the precision with which educational failure can be predicted on the basis of the covariates under study, I think the authors should abstain from drawing clinical conclusions that are not supported by their results.

ANSWER: The wording of the conclusion has been rephrased. Hopefully this change is satisfactory, and better reflect the findings in the current study.

2. In the Discussion, it is suggested that the association between maternal smoking during pregnancy and offspring’s school performance is a casual one. This may definitely be so, but given that maternal IQ and smoking during pregnancy also have been shown to correlate, the association found by the authors could have other explanations. Since correlation does not prove causation, the authors should be more prudent in the interpretation of their data.

ANSWER: We agree with this comment and the interpretation, and the discussion of the results has been rephrased to better reflect the actual results. Additional text has been added to add additional possible explanations.

3. In Table 4, missing values appears to have huge effects on the analysis since 7 out of 18 analysis cannot be performed because of missing data. 

ANSWER: It is not missing data that are the cause of NA. The variables presented are the results from stepwise regression models and NA in this case implies not being included in the modelling (as only statistically significant variables from table 1-3 were considered, separately, for each outcome). In the footnotes to the table, variables included in each of the models are described. However, given the misunderstanding, we have tried to further clarify the models.

How many study subjects are included in the final models? Given the appearance of the table, I would assume this number to be rather low. Have the authors tried to address this problem?

ANSWER: The final models described in table 4, included a total of 670 individuals. This lowered number is due to participants not having participated in each of the follow-ups (i.e., a participant may have chosen to participate in the 3-year follow-up and to not participate in the 12-year follow-up, and then return in the next follow-up). This reduces the validity of using e.g., imputation of missing data. The lowered number of participants is also a reason to why we chose to, initially, model each table separately in order to identify variables of importance in a larger population.

Minor comments:

1. In the abstract, the abbreviation “CBCL” is not explained. For those not in the field, this makes the abstract less comprehensible.

Answer: We agree with the reviewer and thus the abbreviation has been replaced with Child Behavior Checklist.

2. Is it really necessary to introduce the abbreviation GPA?

Answer: Since it is frequently used, we actually think that using GPA instead of grade point average simplifies the reading of the paper.

3. In the methods, second paragraph, the sentence “….yet viewed as a an education one must complete” suggests, according to my interpretation, that upper secondary schooling is more a question of social and peer pressure than about, for example, increasing the individuals chances of finding a job in the future. Even though I agree with the authors to a large extent, I find this phrasing a little unidimensional.

Answer: Thank you for this comment. We see your point and have therefore removed “yet viewed as a an education one must complete” from the sentence as the message regarding societal expectancy is mentioned in the next sentence. 

4. In the Study Population section, “five municipalities in southern Sweden” is a little vague. Is there a thought behind this way of presenting the study population?

Answer: No, there is no thought behind this presentation more than not finding it useful to provide the names of the municipalities. The reason behind choosing these municipalities were that they were adjacent to one another and represented both urban and rural areas, providing a representative sample. This additional information has now been added to the manuscript.

5. In the 3-year follow-up, the number of life events was dichotomized in <=8 and >8. Why this cut off?

Answer: The cutoff of 8 was chosen because it represented the 90th percentile. This information has been added to the manuscript.

6. At the end of the same paragraph, there are minor typos (“percentilen” should be translated into English, and “child” is misspelt)

Answer: Thank you for noticing the typos. These have been changed. The manuscript has now been scrutinized for additional typos.

7. In the 12-year follow-up, there are also cut offs for dichotomization that are not explained.

Answer: Thank you for this comment. The reasoning behind dichotomizing Rosenberg’s scale has now been added to the manuscript. 

8. Table 4. Does the table show all variables included in the model? This is not entirely clear.

Answer: As described in in the first paragraph of the section “Multiple logistic regression models”, only variables that were found to be statistically significant in tables 1-3 were considered in the models. This information can now also be found in the statistics. NA has now been replaced with “Not included” to highlight that these variables were not considered in the model.

---

## [Editor Report · Decision Letter 2]

12 Dec 2022

Predictors of educational failure at 16 and 19 years of age – SESBiC longitudinal study.

PONE-D-21-22136R2

Dear Dr. Bladh,

We’re pleased to inform you that your manuscript has been judged scientifically suitable for publication and will be formally accepted for publication once it meets all outstanding technical requirements.

Kind regards,

Santosh Kumar

Academic Editor

PLOS ONE
---

## [Editor Report · Acceptance letter]

15 Dec 2022

PONE-D-21-22136R2 

Predictors of educational failure at 16 and 19 years of age – SESBiC longitudinal study. 

Dear Dr. Bladh:

I'm pleased to inform you that your manuscript has been deemed suitable for publication in PLOS ONE. Congratulations! Your manuscript is now with our production department. 

Kind regards, 

on behalf of

Dr. Santosh Kumar 

Academic Editor

PLOS ONE